# Initial Evidence of Variation by Ethnicity in the Relationship between Vitamin C Status and Mental States in Young Adults

**DOI:** 10.3390/nu13030792

**Published:** 2021-02-27

**Authors:** Benjamin D. Fletcher, Jayde A. M. Flett, Shay-Ruby Wickham, Juliet M. Pullar, Margreet C. M. Vissers, Tamlin S. Conner

**Affiliations:** 1Department of Psychology, University of Otago, Dunedin 9016, New Zealand; ben.fletcher@postgrad.otago.ac.nz (B.D.F.); shay.wickham@postgrad.otago.ac.nz (S.-R.W.); 2Independent Researcher, Wellington 6011, New Zealand; jayde.flett@gmail.com; 3Centre for Free Radical Research, Department of Pathology and Biomedical Science, University of Otago, Christchurch 8011, New Zealand; juliet.pullar@otago.ac.nz (J.M.P.); margreet.vissers@otago.ac.nz (M.C.M.V.)

**Keywords:** psychology, mental health, well-being, nutrition, micronutrients, healthy adults, ethnicity, Māori, Pasifika, Pacific, Asian

## Abstract

Higher fruit and vegetable intake has been associated with improved mood, greater vitality, and lower stress. Although the nutrients driving these benefits are not specifically identified, one potentially important micronutrient is vitamin C, an important co-factor for the production of peptide hormones, carnitine and neurotransmitters that are involved in regulation of physical energy and mood. The aim of our study was to investigate the cross-sectional relationship between blood plasma vitamin C status and mood, vitality and perceived stress. A sample of 419 university students (aged 18 to 35; 67.8% female) of various ethnicities (49.2% European, 16.2% East Asian, 8.1% Southeast/Other Asian, 9.1% Māori/Pasifika, 11.5% Other) provided a fasting blood sample to determine vitamin C status and completed psychological measures consisting of the Profile of Mood States Short Form (POMS-SF), the vitality subscale of the Rand 36-Item Short Form (SF-36), and the Perceived Stress Scale (PSS). Participants were screened for prescription medication, smoking history, vitamin C supplementation, fruit/juice and vegetable consumption, kiwifruit allergies, excessive alcohol consumption and serious health issues, and provided age, gender, ethnicity, and socioeconomic status information, which served as covariates. There were no significant associations between vitamin C status and the psychological measures for the sample overall. However, associations varied by ethnicity. Among Māori/Pasifika participants, higher vitamin C was associated with greater vitality and lower stress, whereas among Southeast Asian participants, higher vitamin C was associated with greater confusion on the POMS-SF subscale. These novel findings demonstrate potential ethnicity-linked differences in the relationship between vitamin C and mental states. Further research is required to determine whether genetic variation or cultural factors are driving these ethnicity differences.

## 1. Introduction

Higher fruit and vegetable intake is associated with various mental health benefits, including improved mood, vitality, and well-being, as well as decreased depression, anxiety, and stress [1,2,3,4,5,6]. However, the mechanisms by which fruit and vegetable intake confers mental health benefits remain unknown. Higher levels of micronutrients such as vitamin C have been associated with improved mood [1,7,8]. Individuals supplemented with high vitamin C foods, such as kiwifruit or vitamin C tablets, have also reported decreased total mood disturbance and fatigue, and increased levels of well-being and vitality [1,9]. Furthermore, mental health benefits are closely related to changes in vitamin C levels, but not other micronutrient levels, suggesting vitamin C may be an important factor underlying improvements [1,7].

Vitamin C is a required co-factor for a number of enzymes involved in the synthesis of various neurotransmitters, hormones, and catecholamines that may have an impact on mood states [10]. Humans are one of only a few species unable to synthesise vitamin C and, for us, it is therefore a dietary requirement, making us vulnerable to vitamin C deficiency [11]. One of the first signs of vitamin C insufficiency is the onset of lethargy and psychological symptoms, supporting the idea that vitamin C plays a role in mood outcomes. Low vitamin C intake can lead to hypovitaminosis C (<23 µmol/L) and scurvy (<11 µmol/L), a fatal condition characterised by early symptoms of depression, fatigue, and irritability, in addition to poor wound healing and gum disease [10,12,13,14,15,16].

A limited number of studies have specifically assessed the impact of vitamin C on mood in isolation from other micronutrients. Several case studies have reported that vitamin C supplementation in participants with various morbid and co-morbid diseases leads to improvements in measures of depression, fatigue, total mood disturbance, vigour, global health status, and physical, emotional, cognitive and social functioning [8,17,18,19,20,21,22]. However, healthy individuals need to be assessed to determine the generalisability of these associations. Dietary intervention with vitamin C has been reported to result in greater well-being and decreased levels of fatigue and anxiety in healthy participants [9,23,24]. In addition, a cross-sectional study of 139 healthy men (aged 18 to 35) found a small but statistically significant relationship between blood plasma vitamin C levels and mood, measured using the POMS questionnaire [7]. Participants with higher vitamin C levels reported a lower total mood disturbance, which was driven by reduced levels of depression, anger, and confusion on the POMS subscales. However, only men from a relatively homogenous sample of NZ Europeans (76%; Māori (9%); Unspecified (15%)) were assessed, and confounding variables which can influence vitamin C status and mood, such as smoking and alcohol use, were not measured. Several studies have indicated that ethnicity is a factor in variations in the prevalence of mood disturbance, although causation for these differences is by no means established [25,26].

The aim of the current study was to determine the association between blood plasma vitamin C and mood-related outcomes (mood disturbance, vitality, and perceived stress) in a larger and more ethnically heterogeneous sample of healthy adult men and women. We extended previous research by screening out participants with characteristics that might affect, and potentially confound, their vitamin C levels and mood-related outcomes (e.g., smoking, excessive alcohol consumption, prescribed medication, or pre-existing health conditions, such as diabetes [27,28,29,30]) and controlled for demographic and health characteristics related to both mental health and vitamin C outcomes. Questionnaire measures of mood (POMS-short form), vitality (SF-36), and perceived stress (PSS), as well as fasting blood plasma vitamin C levels, were collected. It was hypothesised that vitamin C levels would be positively associated with mood and vitality scores, and negatively associated with perceived stress, when controlling for covariates. Exploratory analyses tested gender and ethnicity as moderators of the relationship between vitamin C and mood, as less is known about these associations in non-European and mixed gendered samples.

## 2. Materials and Methods

### 2.1. Participants and Procedure

Participants were recruited during screening for the KiwiC for Vitality Intervention Study (see Conner et al., 2020)) [9]. The trial was preregistered with the Australian and New Zealand Clinical Trial Registry (Trial ID: ACTRN12617001031358) and approved by the New Zealand Health and Disability Ethics Committee (17/NTB/104). Psychological measures were added to the screening protocol for the second wave of recruitment and these participants were included in this analysis. For full protocol and methods used, see Conner et al. (2020) [9]. Recruitment for the present dataset began in February 2018 and ended in April 2018, and used advertisements placed around the University of Otago and Otago Polytechnic campuses in Dunedin, New Zealand. Students interested in participating signed up through a brief online survey to pre-assess their eligibility (see Table 1). Those who met eligibility criteria were invited to a 30-minute clinic visit in the Department of Human Nutrition clinic located at the University of Otago.

During the clinic visit, participants signed informed consent before completing a one-page re-screening survey asking about health conditions, medication use, fear of needles, fruit and vegetable consumption, vitamin C supplement usage, juice intake, age, and fasting status, to ensure eligibility criteria were still met. Participants then gave a fasted blood sample, after which they completed the rest of the survey detailing measures of demographic characteristics, mood (POMS), vitality (SF-36), and stress (PSS; see Measures section). After completing the survey, participants were provided with breakfast and reimbursed $20.

### 2.2. Measures

#### 2.2.1. Vitamin C Analysis

Participants provided one 9 mL sample of fasting blood, collected into a BD EDTA Vacutainer that was immediately placed on ice and processed at 4 °C within 2 h of collection. Plasma was separated by centrifugation at 1000× *g* for 10 min and 700 μL was extracted with 700 μL of cold 0.54 mol/L perchloric acid/DTPA solution [31]. Samples were vortexed and centrifuged at 13,000× *g* for two minutes to pellet the protein precipitate. Two samples of 500 μL supernatant were stored at −80 °C. Vitamin C was measured by High Performance Liquid Chromatography with electrochemical detection (HPLC-ECD) [31].

#### 2.2.2. Demographics

The survey was used to record age, gender (male, female, gender diverse), ethnicity (based on New Zealand census categories, tick all that apply: New Zealand European, Māori, Samoan, Cook Island Māori, Tongan, Niuean, Chinese, Indian, Another stated ethnicity with free text response), and current socioeconomic status (SES) using three items (“I have enough money to buy the things I want”; “I don’t need to worry too much about paying my bills”; “I don’t think I’ll have to worry about money too much in the future”) rated from 1 (strongly disagree) to 7 (strongly agree) which were averaged (Cronbach’s α = 0.762) [32].

#### 2.2.3. Mood

The Profile of Mood States Short Form (POMS-SF) [33] is a 35-item measure of mood experienced “during the past week, including today”. The items cover six dimensions of mood states: Tension, depression, anger, fatigue, confusion, and vigour and are rated on a five-point Likert scale (1 = not at all to 5 = extremely). The positive items for vigour were summed together and subtracted from the sum of the negative items (tension, depression, anger, fatigue and confusion) to determine an overall mood score, called total mood disturbance (TMD). TMD scores for the POMS short form range from −20 to 100, with higher scores indicative of greater total mood disturbance (Cronbach’s α = 0.899). Subscales were also analysed separately (αs tension = 0.843, depression = 0.825, anger = 0.799, fatigue = 0.860, confusion = 0.773, vigor = 0.828).

#### 2.2.4. Vitality

The Rand 36-Item Short Form (SF-36) [34] is a 36-item patient report survey measuring overall health. The four-item subscale measuring vitality “during the past week including today” was used in the present study (did you feel: Full of life, have a lot of energy, worn out, and tired). Items were rated on a six-point Likert scale (0 = none of the time to 5 = all of the time). Scores were recoded to a 0 to 100 scale (0, 20, 40, 60, 80, 100; reverse scoring worn out and tired), and averaged, with higher scores indicating greater vitality (Cronbach’s α = 0.740).

#### 2.2.5. Stress

The four-item version of the Perceived Stress Scale (PSS) [35] was used to measure the degree to which situations in one’s life are appraised as stressful “during the past week, including today”. Items were rated on a five-point Likert scale (0 = never to 4 = very often). Scores range from 0 to 16, with higher scores indicating higher perceived stress (Cronbach’s α = 0.645).

### 2.3. Analysis

Data were analysed with SPSS (IBM SPSS Statistics, version 26) and R (version 3.6.0) with the alpha level set at 0.05. To determine the relationship between vitamin C levels and demographic factors, one-way ANOVAs were used for categorical variables, independent t-tests were used for binomial variables, and Pearson’s r was used for continuous variables. Initially, scatterplots of the relationship between vitamin C and the three psychological measures (total mood disturbance, vitality, and perceived stress) for the entire sample were created. An unadjusted regression line was fitted for each plot. Multiple regression was then conducted to determine the adjusted association between vitamin C and the three psychological measures controlling for gender (0 = male, 1 = female), age (centred), SES (centred), and ethnicity (dummy coded). To dummy code ethnicity, participants were grouped into five categories (Europeans, East Asians, Southeast and other Asians, Māori and Pasifika, Other). Originally, all participants of Asian descent were grouped together. However, due to finding heterogeneity in the relation between vitamin C and mood outcomes for participants from different Asiatic regions, we subsequently separated East Asian participants from Southeast and Other Asian participants (Table 2). Ethnicity was therefore entered as a set of four dummy codes: Ethnicity D1 compared European vs. East Asians, Ethnicity D2 compared European vs. Southeast and Other Asians, Ethnicity D3 compared European vs. Māori/Pasifika and Ethnicity D4 compared Europeans with all remaining minority ethnicities.

Next, we examined whether the relationship between vitamin C and the outcomes varied by ethnicity. Scatterplots of the relationship between vitamin C and the three psychological measures (total mood disturbance, vitality, and perceived stress) for the five ethnicity groups were created separately and an unadjusted regression line was fitted for each plot. A fully adjusted moderation model was then used to determine whether there were ethnicity differences in the relationship between vitamin C and each outcome, adjusting for the covariates as above. In the fully adjusted moderation model, the interaction terms for vitamin C (centred) by ethnicity dummy codes were added. Initially, Europeans were set as the reference group, with subsequent models changing the reference group to test for all differences between groups. Gender as a moderator was also tested in the fully adjusted moderation model, by entering the vitamin C (centred) by gender (male = 0, female = 1) interaction term.

## 3. Results

### 3.1. Descriptive Statistics

Figure 1 shows the flow diagram of participants. A total of 424 participants attended the clinic visit and signed informed consent, after which two participants were excluded before blood was taken due to taking prescription medication and three participants were unable to give blood. The final sample for analysis was 419 participants (135 males, 284 females) aged between 18- and 35-years-old, with a mean age of 21.13 years (Standard deviation (SD) = 3.33), of various ethnicities (see Table 2). Table 2 presents the sample characteristics. Half of the participants identified as European (49.16%), with others identifying as Asian (30.78%), Māori or Pasifika (9.07%), Indian (5.97%), Middle Eastern (0.95%), African (0.72%), Unspecified (0.48%) or Other/Multiple ethnicities endorsed (2.86%). Asian participants were predominantly either East Asian (Chinese 13.6%) or Southeast Asian (Malaysian 7.6%). Participants average SES score was 4.72 (SD = 1.22), indicating that the sample was not financially stressed and did not feel as though they were deprived of resources at the present moment or in the foreseeable future. Vitamin C scores were normally distributed (range 4.29–118.91 µmol/L) and were consistent with the population average, with a mean of 54.90 µmol/L (SD = 20.19) [36]. One percent of the sample had deficient vitamin C (less than 11 µmol/L), 4.5% had low levels of vitamin C (11–23 µmol/L), 33.4% had inadequate vitamin C levels (23–50 µmol/L), and 61.1% had adequate or saturated levels of vitamin C (50+ µmol/L; Appendix A).

Prior to the main analyses, vitamin C differences by demographic factors and covariates were identified, such that differences in vitamin C were found for gender, ethnicity, age, and fruit and vegetable consumption. Specifically, men had a trend lower average vitamin C level (*M =* 52.55, SD = 18.46) than women (*M =* 56.02, SD = 20.90) t (295.11) = −1.721, *p =* 0.086 (unequal variances). Vitamin C also varied by ethnicity (Appendix A). Vitamin C levels were highest among Māori and Pasifika (*M =* 60.96, SD = 16.80) and Europeans (*M =* 59.35, SD = 19.51), and significantly lower in other ethnicities (*M =* 52.76, SD = 20.77), East Asians (*M =* 51.23, SD = 19.16), and Southeast/Other Asians (*M =* 41.83, SD = 18.62), F (4,414) = 11.49, *p* < 0.001 (Figure 2 and Appendix A). Age was negatively correlated with vitamin C levels (r = −0.133, *p =* 0.006), whereby older participants (up to age 35) had lower vitamin C levels. Fruit and vegetable consumption was also significantly positively correlated with vitamin C level (r = 0.201, *p* < 0.001). There were also ethnicity differences in fruit and vegetable consumption that paralleled the ethnicity differences in vitamin C level. Māori/Pasifika reported the highest fruit and vegetable consumption (*M =* 1.89 serves per day, SD = 1.01), followed by Europeans (*M =* 1.68, SD = 0.86), East Asians, (*M =* 1.29, SD = 0.72), Other (*M =* 1.28, SD = 0.75), and Southeast/Other Asians (*M =* 1.03, SD = 0.59: *Welch’s F* (4,127.68) = 14.00, *p* < 0.001). SES was not significantly correlated with vitamin C level (r = −0.079, *p =* 0.105).

Average total mood disturbance was relatively low in this sample (Mean (*M*) = 9.88, SD = 14.42), with scores above 16 indicative of mood disturbance. Average vitality scores were relatively high (*M =* 60.93, SD = 15.09), whereby a score of 100 represents high energy with no fatigue, and a score of 0 represents no energy and high fatigue. Finally, the average perceived stress scores were relatively low in this sample (*M =* 5.31, SD = 2.06), where the scale ranged from 0 to 16, with higher scores representing higher perceived stress. POMS Total Mood Disturbance, vitality, and perceived stress were correlated (r = −0.604 between POMS TMD and vitality; r = 0.582 between POMS TMD and perceived stress; r = −0.431 between vitality and perceived stress; all *p* < 0.001).

### 3.2. Relationship between Vitamin C and Mood, Vitality, and Stress

#### 3.2.1. Overall Sample

Figure 3 shows scatterplots of the relationship between vitamin C and the three mood-related outcomes for all participants. Vitamin C was not significantly associated with total mood disturbance or vitality for the total sample (Table 3). These patterns did not change in the adjusted models. There was a trend for higher vitamin C being associated with lower perceived stress for the total sample when controlling for gender, age and socioeconomic status (*p* = 0.098). When testing the POMS subscales, vitamin C was not significantly associated with tension, depression, fatigue, confusion or vigour, either as an unadjusted predictor or adjusted (See Table 4). There was a trend for higher vitamin C being associated with less anger (*p =* 0.076), which was maintained when adjusting for covariates (*p =* 0.090).

#### 3.2.2. Moderation by Gender

There were no gender differences in the relationship between vitamin C and any of the psychological outcomes (Appendix A, Vitamin C x Gender coefficients).

#### 3.2.3. Moderation by Ethnicity

There were ethnicity differences in the relationship between vitamin C and vitality and perceived stress, but not POMS total mood disturbance. The general pattern was that higher vitamin C was associated with better outcomes for Māori/Pasifika, but poorer outcomes for Southeast/Other Asians (Figure 4, Figure 5 and Figure 6; Table 3). Among Māori/Pasifika, higher vitamin C was associated with greater vitality (adjusted simple slope b (SE) = 0.373 (0.163), *p =* 0.023; Appendix A), which was significantly different from the pattern found for Southeast/Other Asians (contrast b (SE) = −0.560 (0.175), *p =* 0.001), but not significantly different from the other groups (Table 3; Appendix A). Southeast/Other Asian participants also significantly differed from Europeans (contrast b (SE) = −0.309 (0.116), *p =* 0.008; Appendix A; Appendix A), and Other Ethnicity participants (contrast b (SE) = 0.389 (0.147), *p =* 0.009; Appendix A).

Similar ethnicity differences were found for perceived stress, as shown in Figure 6. Among Māori/Pasifika participants, higher vitamin C was associated with lower perceived stress (adjusted simple slope b (SE) = −0.048 (0.022), *p =* 0.025; Appendix A), which was significantly different from the pattern found for Southeast/Other Asians (contrast b (SE) = 0.053 (0.023), *p =* 0.022), and Europeans (contrast b (SE) = 0.044 (0.020), *p =* 0.031), but not significantly different from the other groups. A similar pattern emerged for the other ethnicity group, in which higher vitamin C was associated with lower perceived stress (adjusted simple slope b (SE) = −0.031 (0.016), *p* = 0.048), although it was not significantly different from patterns for any of the other groups (Appendix A).

There were no ethnicity differences for POMS Total Mood Disturbance (Figure 4; Appendix A), but there were differences in the POMS confusion subscale (Table 4). Among Southeast/Other Asians, higher vitamin C was associated with greater confusion (adjusted simple slope b (SE) = 0.057 (0.027), *p =* 0.031; Appendix A), which was significantly different from the pattern found for Māori/Pasifika (contrast b (SE) = −0.079 (0.037), *p =* 0.034) and the other ethnicity group (adjusted simple slope b (SE) = −0.071 (0.031), *p* = 0.024; Appendix A).

## 4. Discussion

This study is the first to examine the relationship between blood plasma vitamin C levels and mental states by ethnicity. We noted an unexpected variation by ethnicity; to our knowledge there has been no prior report indicating an ethnic difference in the relationship between vitamin C status and mood, vitality and perceived stress. The most notable finding was that only Māori/Pasifika people showed a significant positive association between vitamin C levels and vitality, and a negative association with perceived stress. Higher vitamin C levels for Māori/Pasifika were associated with feeling more “full of life”, energetic and less stressed. In contrast, Southeast Asians and Other Asians tended to report opposing relationships between vitamin C and vitality, perceived stress, and confusion, although the patterns were much weaker than those found for Māori/Pasifika. Whereas the contrast tests between Southeast/Other Asians and Māori/Pasifika were significant due to their opposing patterns, when testing Southeast/Other Asians alone, only the POMS subscale of confusion was statistically significant in adjusted models (with higher vitamin C predicting greater confusion for Southeast/Other Asians).

Average vitamin C status also differed with ethnicity: Māori/Pasifika participants had the highest average plasma vitamin C levels, followed by Europeans, Other ethnicities, East Asians and Southeast Asian/Other Asian participants. Other research has shown that individuals of African or Asian descent tend to have a lower vitamin C status than individuals of European descent [37,38] and a recent review has documented potential regional differences in vitamin C status across the globe [39]. No data were reported for Pacific regions and little data was available for Asian regions. Determining the reasons for the variations in vitamin C status is a challenge, as differences can be due to dietary composition, smoking status, health variables, socioeconomic status and inaccuracies in measurement of vitamin C in plasma samples [31]. In our study, we pre-selected a relatively healthy non-smoking population with similar education levels and socioeconomic status which raises the question of what factors, such as genetic differences or variation in dietary habits, might affect vitamin C requirements and the potential link between vitamin C and mental functioning.

One possible factor affecting vitamin C status is genetic variation affecting gene expression or polymorphisms in proteins involved in vitamin C metabolism within the body. There are polymorphisms in the sodium-dependent vitamin C transport proteins (SVCTs) that regulate uptake and distribution of the vitamin to the tissues and these polymorphisms could affect vitamin C availability throughout the body ascorbate levels [40]. However, little information is available on how this might affect different ethnic groups. Another possibility could be genetic variation in proteins involved in protection against oxidative stress such as haptoglobin and glutathione-S-transferases. Inefficiencies in the detoxification of oxidants could affect ascorbate levels in the body. For example, haptoglobin is present as three main phenotypes, Hp1-1, Hp2-1 and Hp2-2 [41]. Haptoglobin binds to free haemoglobin to prevent oxidative damage from haemoglobin-iron peroxidation and therefore could affect vitamin C levels indirectly by modulating turnover due to oxidative loss [41]. The Hp2-2 isoform binds haemoglobin poorly in comparison to Hp1-1, and is therefore likely to be less effective in limiting haem-induced oxidative stress and potentially decreasing vitamin C levels [41,42,43,44]. The Hp2-1 phenotype is a combination of Hp1 and Hp2 alleles, which is an intermediate step between the opposing functional properties of Hp1-1 and Hp2-2 [45]. Simply put, vitamin C oxidation likely occurs at a higher rate in individuals with the Hp2-2 phenotype than Hp2-1 and Hp1-1 phenotypes, contributing to a lowering of vitamin C status [41]. Māori/Pasifika, who are more likely to possess the Hp1 allele [46], were found to generally have higher plasma status, which is consistent with a potentially decreased oxidative stress load. More detailed studies involving a direct correlation of the ascorbate status and Hp phenotype would be required to test this hypothesis.

Whether the reported mental states are associated with Hp phenotype is unknown. Previously, Hp phenotype variations have been associated with psychological disorders, including depression, schizophrenia, psychoses and familial epilepsy [45], but whether Hp phenotype changes are responsible for ethnic differences in the relationship between vitamin C and mental states is unknown.

Another possible explanation driving the ethnicity differences in our study could be variations in the availability of other micronutrients, in addition to vitamin C, that have a role in brain functioning and mood. The current study did not incorporate other dietary measures, such as diet records or biomarkers, to assess other micronutrients or diet types as possible confounding factors. For example, a Japanese diet, categorised by higher consumption of soy and fruit and vegetables has been associated with reduced depression [47]. In contrast, so called “Western Diets”, which consist of refined grains, high sugar intake, processed foods, beer, and fried foods, have been linked to poorer mental health [48,49]. More specifically, B vitamins are associated with reduced stress levels and vitamin D has been associated with lower levels of depression and fatigue [50,51]. Cultural variation in diet and food preparation may affect the variation in micronutrients that are typically consumed, which may account for some of the ethnicity linked mental health differences.

In the current study, we did not find a significant relationship between vitamin C status and any mood outcomes for Europeans, which contrasts with the cross-sectional study by Pullar and colleagues [7]. This may reflect differences in sample characteristics between these studies, either driven by differences in sample size (*n =* 419 in our study vs. *n =* 139) or exclusion criteria. In the current study the exclusion criteria included smoking, excessive alcohol consumption, prescription medication and pre-existing health conditions, which can all impact mental health and vitamin C status. For example, individuals who smoke tend to have lower levels of vitamin C than individuals who do not smoke [28]. In addition, smoking has been associated with poorer mental health outcomes, such as higher levels of depression, anxiety, and stress [52,53]. Thus, some previous vitamin C studies may have found a relationship between vitamin C and mental health outcomes which was driven by other health factors rather than a direct effect of vitamin C on mental health. It is also possible that our stringent exclusion criteria made our sample relatively less representative than if we had not screened for these factors. However, the fact that we had a wide range of vitamin C levels that mirrored population estimates makes this less of an issue.

More generally, the current study found no association between mental health and vitamin C status for the total sample. These findings support an earlier New Zealand based cross-sectional study, which addressed a wide range of covariates in 50-year-old adults, suggesting that these findings may be applicable regardless of age [54]. Although, the sample in the earlier cross-sectional study consisted largely of NZ Europeans (83.7%), which may have driven the null result. It may be possible that variations by ethnicity in the relationship between mental health and vitamin C are inadvertently missed, or masked, when samples are homogenous.

Previously, we have indicated in an 8-week randomised controlled trial, that Asian ethnicity may play a role in the relationship between vitamin C and mood over time in response to a vitamin C or kiwifruit intervention [9]. However, one limitation of the current study is that the English versions of the measures have not been verified for all ethnicities. It is possible that interpretations of items varied between ethnic groups, which could influence the current findings. Nevertheless, English comprehension was assumed to be satisfactory because all participants in the current study were enrolled at university in an English-speaking country, all participants indicated understanding of the study (as per the information and consent form, and interactions with the research assistant), and no participants required an interpreter. In addition, it has also been suggested that the POMS-SF can be generalised across ethnic groups [55]. The English version of SF-36 has been validated in many European populations, as well as NZ Europeans, younger Māori (under 45 years of age), and Asian participants (with lupus) [56,57,58]. However, although Pacifika populations may interpret vitality items of the SF-36 to be more representative of mental health than physical health [56], the taxonomy of these items are more reflective of mental health [59]. The PSS has been validated in many countries and languages; however, the use of the English version in a diverse sample may require further investigation [60]. In the current study, all Cronbach alpha levels were acceptable for the respective measures. Thus, ethnic differences in the relationship between vitamin C and, which may be worthy of follow-up.

The current study was strengthened by including a large sample size, an objective measurement of blood vitamin C levels rather than dietary estimates, an ethnically diverse sample, and strict exclusion criteria to eliminate potential confounding factors. Limitations included the cross-sectional correlational design, which limits drawing of causal inferences. We also recruited young adult English speaking university students who consumed fewer than five servings of fruit and vegetables per day, which reduces the generalisability of these findings. However, the average vitamin C level of our sample was consistent with the population average [36], and vitamin C levels in the current sample ranged from deficient to well beyond saturated (4.29–118.91µmol/L), suggesting the sample was representative. To improve generalisability, future research needs to be conducted in older adults of various ethnicities, possibly in different countries or environments with specific measures that are verified for each population of interest. Additionally, in the current study, we did not assess BMI. Higher BMI scores are associated with lower vitamin C [61]. Interestingly, New Zealand Māori and Pasifika populations tend towards having a higher BMI, which would be expected to result in lower vitamin C levels [62]. Asian adults tend to have the lowest rates of obesity, which would be expected to be associated with higher vitamin C levels [62]. However, this was not evident in the current study, which may suggest that BMI was not a driving factor of vitamin C levels. Food preparation methods are also likely to influence vitamin C uptake, as cooking destroys the vitamin. Our study was not intended to analyse the contribution of culinary styles to vitamin C status. However, future studies should consider including additional health measures, such as BMI to control for possible confounds. We were also limited by the number of participants in each ethnic subgroup (Table 2); as ethnicity differences were unexpected, this was not a factor in recruitment. Subsequent studies may need to focus on larger samples of specific ethnicities to thoroughly assess the relationship between ethnicity, vitamin C and mental states.

## 5. Conclusions

To our knowledge, this is the first study to indicate that there may be opposing relationships between vitamin C levels and vitality and perceived stress for different ethnicities. Understanding this relationship could have implications for dietary recommendations to improve mental health and well-being. The differences in vitality and perceived stress predicted from vitamin C levels between ethnic groups present a complex interaction, which may be a result of a differences in dietary habits or differing combinations of micronutrients, rather than vitamin C alone. Alternatively, the relationship between vitamin C and mental health may be moderated by genetic factors, which may indicate differences in the ideal level of vitamin C for optimal mood and mental health for different ethnicities. Further research is needed to replicate and explore the ethnicity differences in the relationship between vitamin C and mood and mental health.

## Figures and Tables

**Figure 1 nutrients-13-00792-f001:**
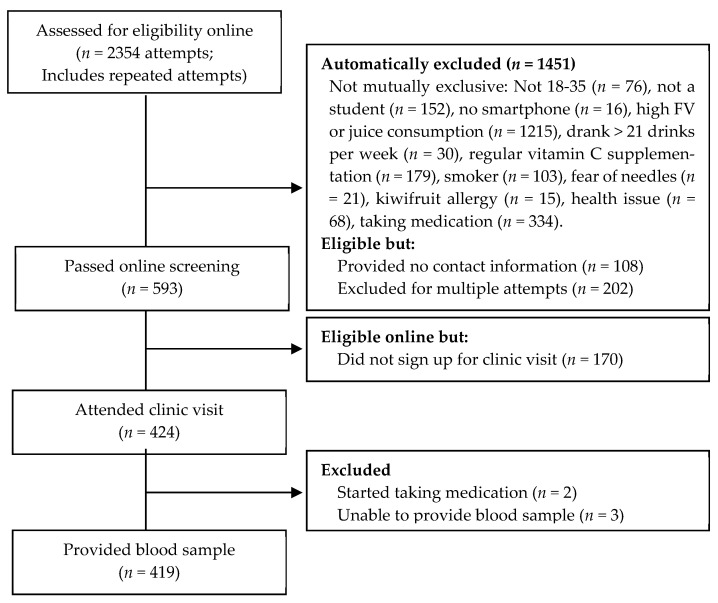
Participant flow diagram.

**Figure 2 nutrients-13-00792-f002:**
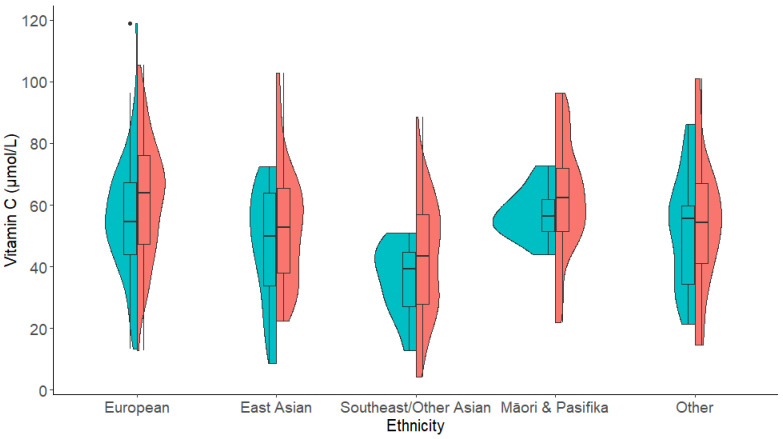
Violin plot showing the vitamin C levels for each ethnicity by gender (male = blue (left) and female = red (right)). Overlaid boxplots show median vitamin C levels for each gender for each ethnicity, interquartile range, and any outliers. See Appendix A for the mean plasma vitamin C levels for each ethnicity by gender subgroup. Europeans (*n* = 206, 80 male, 126 female), East Asians (*n* = 68, 17 male, 51 female), Southeast/Other Asian (*n* = 61, 9 male, 52 female), Māori & Pasifika (*n* = 38, 10 male, 28 female), other ethnicities (*n* = 46, 19 male, 27 female).

**Figure 3 nutrients-13-00792-f003:**
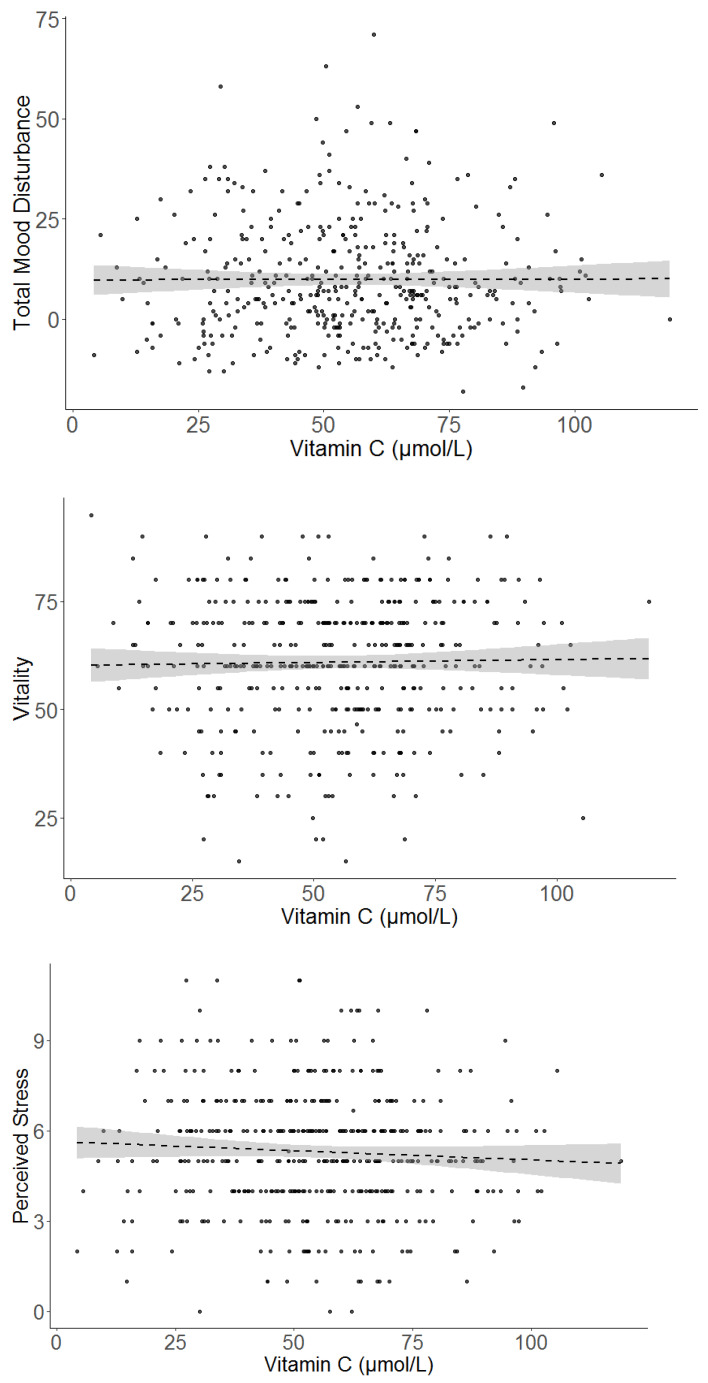
Simple scatterplots of the unadjusted relationship between vitamin C and the three main outcomes (POMS Total Mood Disturbance, vitality and perceived stress) for the sample overall (*n =* 419). Dashed line indicates the line of best fit. Grey shading indicates 95% confidence intervals.

**Figure 4 nutrients-13-00792-f004:**
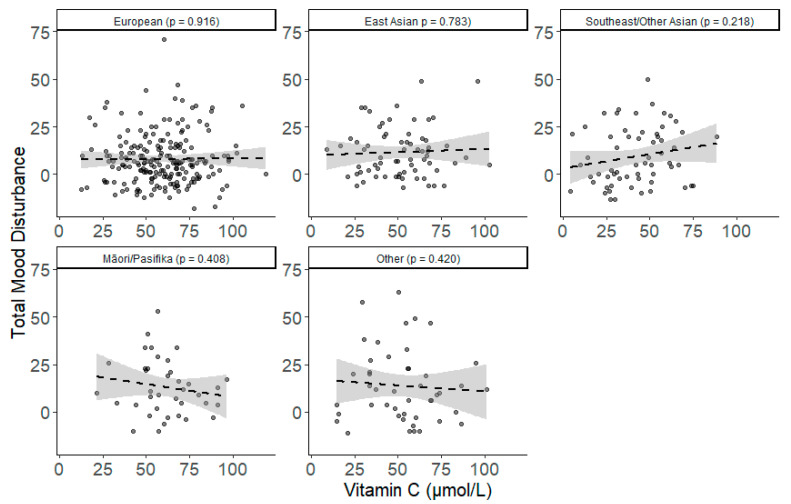
Simple scatterplots of the unadjusted relationship between vitamin C and Profile of Mood States (POMS) Total Mood Disturbance between different ethnic groups (Europeans, East Asians, Southeast/Other Asians, Māori/Pasifika and Other). Dashed line indicates the line of best fit. Grey shading indicates 95% confidence intervals. *p* values for fully adjusted simple slopes.

**Figure 5 nutrients-13-00792-f005:**
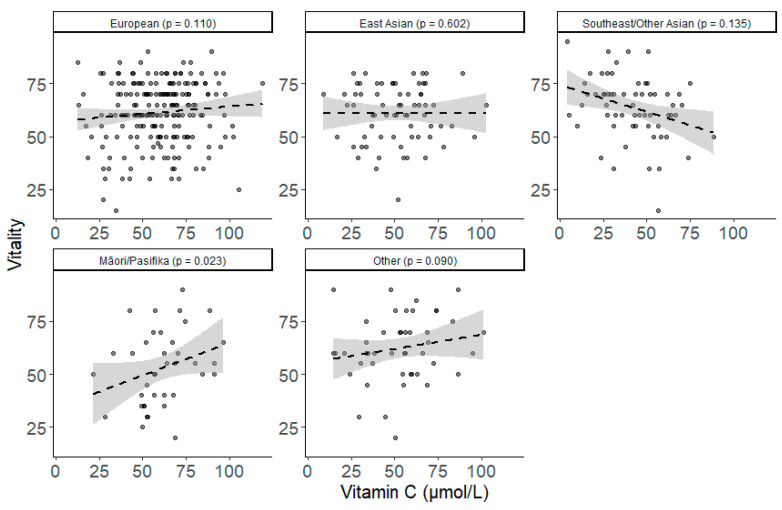
Simple scatterplots of the unadjusted relationship between vitamin C and vitality between different ethnic groups (Europeans, East Asians, Southeast/Other Asians, Māori/Pasifika and Other). Dashed line indicates the line of best fit. Grey shading indicates 95% confidence intervals. *p* values for fully adjusted simple slopes.

**Figure 6 nutrients-13-00792-f006:**
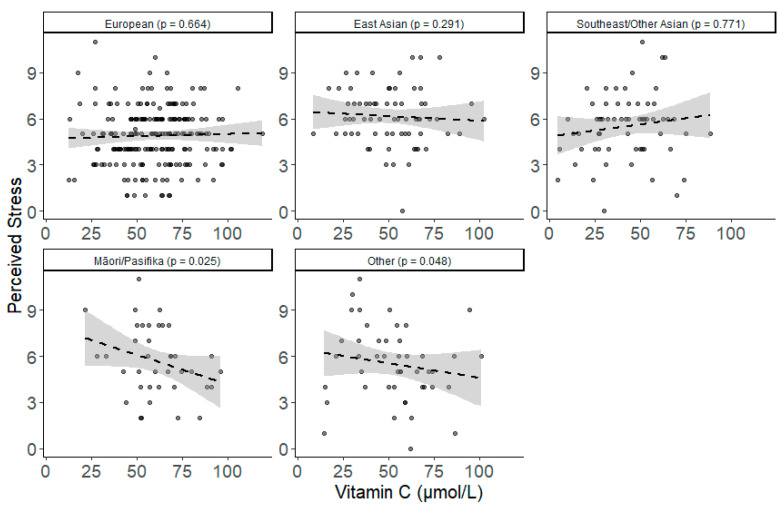
Simple scatterplots of the unadjusted relationship between vitamin C and perceived stress between different ethnic groups (Europeans, East Asians, Southeast/Other Asians, Māori/Pasifika and Other). Dashed line indicates the line of best fit. Grey shading indicates 95% confidence intervals. *p* values for fully adjusted simple slopes.

**Table 1 nutrients-13-00792-t001:** Eligibility criteria assessed in the online screening and screening appointment.

Inclusion Criteria (All Required for Inclusion)	Exclusion Criteria (Only One Required for Exclusion)
Any gender aged 18–35 years	Taking prescription medication (within past three months)
Non-smoker	Allergy/intolerance to kiwifruit
Currently a student	Recent smoker (within past year)
	Taking vitamin C supplements (within past three months)
	High fruit/juice and vegetable consumption (≥5 servings/day)
	Excessive alcohol consumption (>21 standard drinks/week)
	Serious health issues, such as diabetes mellitus, kidney disease, bleeding disorders, or clinical depression
	Fainting due to fear of needles

**Table 2 nutrients-13-00792-t002:** Sample characteristics (*n* = 419).

	**Mean (SD)**		**Minimum**		**Maximum**
Age	21.13 (3.33)		18.00		35.00
SES	4.72 (1.22)		1.00		7.00
Vitamin C in µmol/L	54.90 (20.19)		4.29		118.91
Total mood disturbance	9.88 (14.42)		−18.00		71.00
Vitality	60.93 (15.09)		15.00		95.00
Perceived stress	5.31 (2.06)		0.00		11.00
				***n* (% of sample)**	
Gender		Female		284 (67.78%)	
		Male		135 (32.22%)	
		Gender diverse		0 (0.00%)	
Ethnicity		European		206 (49.16%)	
		NZ European		187 (44.63%)	
		Other European		19 (4.53%)	
		Asian		129 (30.78%)	
		East Asian ^a^		68 (16.23%)	
		Southeast Asian ^b^		34 (8.11%)	
		Other Asian ^c^		27 (6.44%)	
		Māori or Pasifika ^d^		38 (9.07%)	
		Other Ethnicities		46 (10.98%)	
		Indian		25 (5.97%)	
		Multiple Ethnicities		12 (2.86%)	
		Middle Eastern		4 (0.95%)	
		African		3 (0.72%)	
		Not Specified		2 (0.48%)	

SES = socioeconomic status ^a^ Includes Chinese, Hong Kongese, Korean, Japanese and Taiwanese ethnicities. ^b^ Includes Malaysian, Thai, and Singaporean ethnicities. ^c^ Includes Vietnamese, Bangladeshi, Indonesian, Pakistani, Filipino, Siberian, Sri Lankan, mixed Asian and not specified Asian ethnicities. ^d^ Includes Māori, Samoan, Tokelauan, Tongan, Tuvaluan, Fijian, mixed Māori and European (*n =* 21) and mixed Samoan and European (*n =* 3).

**Table 3 nutrients-13-00792-t003:** Stress for the Total Sample and for Different Ethnicity Groups, Unadjusted for Covariates (top), and Adjusted for Covariates (middle) and Adjusted for Covariates and Moderators (bottom).

	Total Mood Disturbance	Vitality	Perceived Stress
**Unadjusted associations**	**b (SE)**	**b (SE)**	**b (SE)**
Total Sample	0.002 (0.035)	0.013 (0.037)	−0.006 (0.005)
European	0.006 (0.047)	0.071 (0.052)	0.003 (0.007)
East Asian	0.033 (0.084)	0.001 (0.086)	−0.006 (0.012)
Southeast/Other Asian	0.150 (0.104)	−0.254 (0.099) *	0.016 (0.015)
Māori/Pasifika	−0.138 (0.144)	0.309 (0.167) †	−0.039 (0.021) †
Other	−0.066 (0.137)	0.135 (0.112)	−0.019 (0.017)
**Adjusted associations ^1^**	**b (SE)**	**b (SE)**	**b (SE)**
Total Sample	−0.010 (0.035)	0.031 (0.037)	−0.008 (0.005) †
European	0.019 (0.048)	0.070 (0.053)	0.004 (0.007)
East Asian	0.022 (0.080)	0.002 (0.086)	−0.009 (0.011)
Southeast/Other Asian	0.119 (0.097)	−0.223 (0.095) *	0.012 (0.014)
Māori/Pasifika	−0.138 (0.151)	0.318 (0.174) †	−0.042 (0.021) †
Other	−0.114 (0.130)	0.151 (0.116)	−0.026 (0.016)
**Adjusted + Moderators ^2^**	**b (SE)**	**b (SE)**	**b (SE)**
European	0.008 (0.073)	0.121 (0.076)	−0.004 (0.010)
East Asian	0.029 (0.107)	0.058 (0.111)	−0.016 (0.015)
Southeast/Other Asian	0.149 (0.121)	−0.188 (0.125)	0.005 (0.017)
Māori/Pasifika	−0.130 (0.157)	0.373 (0.163) *	−0.048 (0.022) *
Other	−0.092 (0.140)	0.201 (0.118) †	−0.031 (0.016) *

b (SE) = unstandardised coefficient (Standard Error), † *p* <0.10; * *p* <0.05. ^1^ Different ethnic groups adjusted for gender, age, and socioeconomic status. ^2^ Full model, adjusted for gender, age, socioeconomic status, ethnicity, gender by vitamin C, and ethnicity by vitamin C interactions. See Appendix A for full models.

**Table 4 nutrients-13-00792-t004:** Associations between Vitamin C and the Profile of Mood Scale (POMS) subscales for the Total Sample and for Different Ethnicity Groups, Unadjusted for Covariates (top), and Adjusted for Covariates (middle) and Adjusted for Covariates and Moderators (bottom).

	Tension	Depression	Anger	Fatigue	Confusion	Vigor
**Unadjusted associations**	**b (SE)**	**b (SE)**	**b (SE)**	**b (SE)**	**b (SE)**	**b (SE)**
Total Sample	0.004 (0.008)	−0.006 (0.006)	−0.012 (0.007) †	0.013 (0.009)	0.001 (0.008)	−0.001 (0.009)
European	0.018 (0.012)	0.005 (0.008)	−0.011 (0.008)	0.000 (0.012)	0.001 (0.011)	0.007 (0.012)
East Asian	−0.012 (0.020)	−0.006 (0.018)	0.005 (0.017)	0.011 (0.021)	0.007 (0.020)	−0.029 (0.022)
Southeast/Other Asian	0.028 (0.023)	0.001 (0.019)	−0.007 (0.020)	0.061 (0.032) †	0.046 (0.022) *	−0.021 (0.026)
Māori/Pasifika	−0.043 (0.035)	−0.007 (0.021)	−0.007 (0.029)	−0.017 (0.042)	−0.037 (0.031)	0.027 (0.036)
Other	0.012 (0.031)	−0.010 (0.025)	−0.014 (0.027)	0.010 (0.029)	−0.014 (0.029)	0.049 (0.027) †
**Adjusted associations ^1^**	**b (SE)**	**b (SE)**	**b (SE)**	**b (SE)**	**b (SE)**	**b (SE)**
Total Sample	0.002 (0.008)	−0.006 (0.006)	−0.011 (0.007) †	0.010 (0.009)	−0.002 (0.008)	0.003 (0.009)
European	0.021 (0.012) †	0.009 (0.008)	−0.008 (0.008)	0.001 (0.013)	0.004 (0.011)	0.008 (0.013)
East Asian	−0.015 (0.019)	−0.008 (0.017)	0.003 (0.017)	0.010 (0.022)	0.004 (0.018)	−0.027 (0.022)
Southeast/Other Asian	0.021 (0.022)	−0.003 (0.018)	−0.009 (0.020)	0.054 (0.032) †	0.043 (0.022) †	−0.012 (0.026)
Māori/Pasifika	−0.037 (0.036)	−0.007 (0.022)	−0.014 (0.029)	−0.022 (0.044)	−0.031 (0.032)	0.027 (0.038)
Other	−0.003 (0.030)	−0.016 (0.024)	−0.017 (0.027)	0.005 (0.029)	−0.029 (0.027)	0.055 (0.028) †
**Adjusted + Moderators ^2^**	**b (SE)**	**b (SE)**	**b (SE)**	**b (SE)**	**b (SE)**	**b (SE)**
European	0.026 (0.018)	−0.002 (0.013)	−0.012 (0.014)	−0.008 (0.020)	0.011 (0.016)	0.006 (0.018)
East Asian	−0.005 (0.026)	−0.016 (0.019)	0.002 (0.020)	0.001 (0.029)	0.016 (0.024)	−0.030 (0.027)
Southeast/Other Asian	0.037 (0.029)	−0.010 (0.022)	−0.011 (0.023)	0.051 (0.032)	0.057 (0.027) *	−0.025 (0.030)
Māori/Pasifika	−0.031 (0.038)	−0.018 (0.028)	−0.010 (0.030)	−0.027 (0.042)	−0.021 (0.034)	0.022 (0.040)
Other	0.013 (0.028)	−0.022 (0.021)	−0.018 (0.022)	−0.002 (0.031)	−0.013 (0.025)	0.050 (0.029) †

b (SE) = unstandardised coefficient (Standard Error), † *p* <0.10; * *p* <0.05. ^1^ Adjusted for gender, age, and socioeconomic status. ^2^ Full model, adjusted for gender, age, socioeconomic status, ethnicity, gender by vitamin C, and ethnicity by vitamin C interactions, see Appendix A.

## Data Availability

The data presented in this study are available on request from the corresponding author. The data are not publicly available due to privacy and ethical reasons.

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
