# Peer review of "Initial Evidence of Variation by Ethnicity in the Relationship between Vitamin C Status and Mental States in Young Adults"

_nutrients, 2021, doi:10.3390/nu13030792_

Round 1
Reviewer 1 Report
Overall, the study was well thought out with a very novel idea that adds some vital knowledge to vitamin C and mental health. The manuscript is written well and the tables and figures are displayed very well.
In the introduction, I would recommend adding some brief information on the already known differences in plasma vitamin C concentrations between ethnicities. Some brief information on differences in mood states between ethnicities may also add to this.
The sample consisted of a very young adult cohort, more likely to display optimal vitamin C levels and mood states. It is well known that older adults are more vulnerable to experience states of vitamin C deficiency. Based on the current study results, it may be difficult to confer the exact relationship between plasma vitamin C and mood in adults without including a cohort over the age of 60. Within the limitations/future directions section, I would highly recommend that trials need to be performed on adults over the ages of 60 to better determine this relationship.
I would also consider changing the title to ‘young adults.’
In the discussion, there are a number of additional explanations which could account for the observed variations in vitamin C concentrations. Culinary styles and methods of food preparation may substantially reduce the vitamin C levels. Cooking of vegetables and fruits can reduce their vitamin C content. There may be link between plasma vitamin C and HDL cholesterol. Decreased levels of HDL cholesterol in South east Asians may influence increased blood utilization of vitamin C due to increased oxidative stress.
I would recommend adding a further explanation for the results observed amongst the Southeast/Other Asian sample, particularly given that vitamin C appeared to worsen mood and vitality in this sample. It may have something to do with hormones, norepinephrine, gut health, etc.
The discussion highlights that dietary patterns were not measured. I would recommend mentioning individual nutritional biomarkers which are potentially associated with mood and may display different levels between ethnicities, such as Vitamin B 12 and vitamin D.
Additional confounding factors that may be considered include household size, educational status, environmental pollution and supplementation of other supplements that may interact with plasma vitamin C levels and/or mood.
The discussion points out that ‘all participants in the current study were enrolled at university in an English-speaking country.’ This is a major point which may impact the generalisability of the results. There still remains the question of whether participants from certain ethnicities still living in their natural habitats would produce the same results. Future studies could aim to gather data directly from participants living in countries/regions representative of their ethnicities to better grasp a sense of vitamin C level and mood state.
Minor grammatical error in the conclusion, remove the letter ‘a’ before differences in dietary habits.
Reviewer 2 Report
This is a well-written paper describing a well-designed study of the relationship between blood vitamin C status and several mood and mental states. The analysis is thorough. I appreciate the presentation of scatterplots, which allow the reader to see clearly the nature of the data. I particularly appreciate the fact that a number of potential confounding factors were excluded by design and exclusion criteria, such as smoking status and supplement use. Of course, that introduces its own set of confounding factors or issues for interpretation (the youth of the sample, and that they were all university students), of which there could be more discussion.
Some individual comments and suggestions:
Section 2.1, Participants. I know you mentioned the number of subjects in the abstract, but I like to see it in this descriptive section.
Section 2.3, Analysis. In the last paragraph, in which you describe the models, please insert the reference to the Supplementary tables which show the models.
Section 3.1, Descriptive statistics. I would like to see a table in the Supplementary tables which shows number of participants by ethnicity and gender. Figure 2 makes clear that some of the ethnic-gender subgroups are very small.
Also somewhere, perhaps here, I would like to see a better description of the Vitamin C levels in the ethnic groups. For example, Figure 2 indicates that one quarter of Southeast Asians have blood vitamin C levels that are very low, c. 27 umol/L, which is down in the marginal adequacy level. On the other hand, all Maori males had blood levels higher than the highest of the Southeast Asians. (I would like to see a table in Supplementary tables showing such data.)
Also, are body weight and BMI available in the data? Both can influence blood vitamin C level. Are Maoris notably thinner than the other groups? What about overweight, which causes oxidative stress? There are reports of a growing overweight problems among southeast Asians. Perhaps in the Supplementary tables?
Figure 2. I really like this figure. I would like to see the numbers of subjects for each ethnic group (and gender?). For example, I figured from Table 2 that in the Maori/Pasifika group there are only n=38. It would be helpful if this were in the figure or label.
Section 4, Discussion. I would like to see more acknowledgement of the fact that overall there was no association between Vitamin C level and mental factors, contrary to what others have found. And more acknowledgement of the fact that although ethnic differences were found, they were in very small samples which were marked by striking differences in blood levels. And they could have been simply chance.
Very good paragraph on genetic factors.
